# Racemic Norlignans as Diastereoisomers from *Ferula sinkiangensis* Resins with Antitumor and Wound-Healing Promotion Activities

**DOI:** 10.3390/molecules27123907

**Published:** 2022-06-17

**Authors:** Ying-Shi Li, Bao-Chen Yang, Shu-Min Zheng, Yong-Xian Cheng, Hong-Hua Cui

**Affiliations:** 1School of Traditional Chinese Medicine, Guangdong Pharmaceutical University, Guangzhou 510006, China; lws1030@163.com; 2Institute for Inheritance-Based Innovation of Chinese Medicine, School of Pharmaceutical Sciences, Health Science Center, Shenzhen University, Shenzhen 518060, China; ybc15801655293@hotmail.com (B.-C.Y.); zheng80231223@163.com (S.-M.Z.); 3Guangdong Key Laboratory of Functional Substances in Medicinal Edible Resources and Healthcare Products, School of Life Sciences and Food Engineering, Hanshan Normal University, Chaozhou 521041, China

**Keywords:** *Ferula sinkiangensis*, Apiaceae, plant resins, ferulasinkins, norlignans, antitumor, wound healing

## Abstract

Ferulasinkins A–D (**1**–**4**), four new norlignans, were isolated from the resins of *Ferula sinkiangensis*, a medicinal plant of the Apiaceae family. All of them were obtained as racemic mixtures, chiral HPLC was used to produce their (+)- and (−)-antipodes. The structures of these new compounds, including their absolute configurations, were elucidated by spectroscopic and computational methods. This isolation provides new insight into the chemical profiling of *F. sinkiangensis* resins beyond the well-investigated structure types such as sesquiterpene coumarins and disulfides. Compounds **2a** and **3a** were found to significantly inhibit the invasion and migration of triple-negative breast cancer (TNBC) cell lines via CCK-8 assay. On the other hand, the wound-healing assay also demonstrated that compounds **4a** and **4b** could promote the proliferation of human umbilical vein endothelial cells (HUVECs). Notably, the promoting effects of **4a** and **4b** were observed as more significant versus a positive control using basic fibroblast growth factor (bFGF).

## 1. Introduction

The genus *Ferula*, belonging to the Apiaceae family, includes more than 150 species that are widespread throughout the Mediterranean area and Central Asia, such as Turkey, Iran, Pakistan, and northwest China [1]. Most of the *Ferula* plants have a pungent odor owing to the presence of disulfides [2]. Modern pharmacological studies have established the anti-ulcer [3], antibacterial [4], anti-inflammatory [5], antitumor [6], and antiviral activities [7] of this genus.

*Ferula sinkiangensis*, an endemic specie of the *Ferula* in China, is mainly distributed in the Xinjiang Uygur Autonomous Region. The resins produced by *F. sinkiangensis* are recorded in the Chinese Pharmacopoeia as a kind of traditional Chinese medicine called “A Wei” [8]. As a kind of folk medicine, it is often used to reduce the symptoms of indigestion, lumps, ovarian cysts, joint pain, and wound infection by Uygur people [9]. Previous phytochemical investigations of this plant mainly led to the isolation of ferulic acid derivatives [10], phenylpropanoid derivatives [11], sesquiterpene coumarins [12], steroidal esters [13], and disulfides [14]. However, the constituents of lignans have rarely been reported in recent years.

Plant resins have become our research interest in recent years. As a result, several structure intriguing compounds have been described by us from resins with different sources [15,16,17]. Considering the medicinal importance of *F. sinkiangensis* resins, an investigation of the EtOAc extracts of this material was initiated and four new norlignans that possess tetrahydrofuran rings were isolated. Differing from ordinary lignans, the chemical structures of norlignans are composed of phenylpropane (C_6_–C_3_) and phenylethane (C_6_–C_2_) units [18]. Some norlignans have been established that they possess antitumor, anti-oxidant, and anti-viral activities [19,20]. Since norlignans have not been reported in *Ferula* plants, it greatly stimulates our interest to initiate an investigation on the antitumor and wound-healing properties of these compounds based of their traditional medicinal functions. The isolation, structure identification, and biological evaluation of these four compounds are described below.

## 2. Results and Discussion

### 2.1. Structure Elucidation of the Compounds

An EtOH extract of the resins of *F. sinkiangensis* was partitioned between H_2_O and EtOAc. The EtOAc portion was subjected to a combination of column chromatography including MCI gel CHP 20P, RP-18, YMC gel ODS-A-HG, silica gel, Sephadex LH-20, preparative thin-layer chromatography, and semipreparative HPLC to afford four new norlignans (Figure 1).

Compound **1** was initially isolated as a colorless gum, and the molecular formula C_19_H_22_O_7_ was confirmed by HR-ESI-MS ([M − H]^−^, *m*/*z* 361.1293; calcd for C_19_H_21_O_7_, 361.1278) with nine degrees of unsaturation. In the ^1^H NMR spectrum (Table 1), six aromatic protons (*δ*_H_ 7.01, 6.99, 6.89, 6.84, 6.78, and 6.75), two methoxy groups (*δ*_H_ 3.86 and 3.84), four methine protons (*δ*_H_ 4.68, 4.61, 4.40, and 4.02), and two methylene protons (*δ*_H_ 1.91 and 1.57) are observed. Multiplets and coupling constants of these aromatic protons suggest the presence of two 1,3,4-trisubstituted phenyl groups [21,22]. The ^13^C NMR and DEPT spectra (Table 1) show two aromatic rings with carbon signals for ten methines, one methylene, two methoxyls, and six nonprotonated carbons. These NMR data indicate that **1** might be a lignan derivative with an extra ring corresponding to the extra degree of unsaturation except for two phenyl rings accounting for eight degrees of unsaturation. The structure construction of **1** was mainly performed by 2D NMR experiments (Figure 2). The ^1^H-^1^H COSY spectrum exhibits correlations of H-7/H-8/Ha-9/Hb-9 and Hb-9/H-8′/H-7′, in consideration of the HMBC correlations of H-7′/C-8, Ha-9/C-7′ and the chemical shifts of C-7′ (*δ*_C_ 89.8), C-8′ (*δ*_C_ 79.2), C-8 (*δ*_C_ 83.8), indicate the presence of a tetrahydrofuran skeleton with oxygenated C-8′. Moreover, the ^1^H-^1^H COSY cross-peaks of H-5/H-6 and H-5′/H-6′ demonstrate that these four protons are nonsubstituted, confirming the absence of those two 1,3,4-trisubstituted phenyl groups. Further HMBC correlations of H-7′/C-1′, C-2′, and C-6′ indicate that a benzene ring was connected to this tetrahydrofuran ring via C-7′. On the other hand, HMBC correlations of H-7/C-1, C-2, C-6, C-8, and H-8/C-1 allow us to deduce that another benzene ring is linked to C-7. The chemical shifts of C-3, C-4, C-3′, and C-4′ indicate their oxygen-bearing nature. As for the position of methoxy groups at benzene rings, the ROESY experiment can help provide evidence. The HMBC cross-peaks of 3-OCH_3_/C-2, C-3, C-4, and 3′-OCH_3_/C-2′, C-3′, C-4′, and ROESY correlations of 3-OCH_3_/H-2 and 3′-OCH_3_/H-2′ allow us to assign the position of methoxy and hydroxy groups in each benzene ring. In addition, the chemical shift for C-7 (*δ*_C_ 78.1) indicates that it is oxygenated by a hydroxy group. Thus, the planar structure of compound **1** was deduced (Figure 1) and named ferulasinkin A.

The relative configuration of compound **1** was determined by coupling constants and the interpretation of ROESY experiments. Cross-peaks of H-8/H-7′ and H-8/H-8′ suggested that H-8, H-7′, and H-8′ were all in the same direction (Figure 2). As for the stereochemistry at C-7, it has been reported that ^3^*J*_HH_ values of acyclic vicinal diol groups seemingly follow an empirical rule: a relatively larger value (more than 6.0 Hz) corresponds to a *threo* configuration, whereas a smaller value (less than 5.0 Hz) corresponds to an *erythro* configuration [23,24]. Therefore, the coupling constant between H-7 and H-8 (*J*_7,8_ = 7.0 Hz) allows us to assign the *threo* configuration.

In general, natural compounds are enzyme-catalyzed products with chiral centers. Interestingly, chiral HPLC analysis indicates that compound **1** contains its enantiomers, corresponding to a small value of optical rotation which might result from the measurement errors. Thus, chiral separation of **1** was conducted to afford **1a** and **1b** (Appendix A), whose absolute configurations were clarified by electronic circular dichroism (ECD) calculations. It was found that the experimental ECD curve of **1a** agrees well with the calculated one of (7*R*, 8*R*, 7′*R*, 8′*R*)-**1** (Figure 3), allowing to assign the absolute configuration of **1a**. As a result, the structure of **1** was finally assigned and named (+)-ferulasinkin A and (−)-ferulasinkin A, respectively.

Detailed analysis of the ^1^H, ^13^C NMR, ^1^H-^1^H COSY and HMBC spectra of compounds **1**–**4** (Table 1 and Table 2, Figure 2), found their extreme similarities, demonstrating that all of them share the same planar structure (Figure 1).

Compound **2** was also obtained as a colorless gum. ROESY enhancements of **2** were observed between H-7/H-7′, H-8′, and H-8/H-6′ (Figure 2), which indicates that H-7′ and H-8′ are co-facial, while H-8 is on the other face. The relative configuration of H-7 and H-8 was supposed to be *threo* by the coupling constant of H-7 and H-8 (*J*_7,8_ = 7.7 Hz). Compound **2** was also isolated as a racemic mixture. Subsequent chiral separation on HPLC afforded its antipodes, **2a,** and **2b** (Appendix A). The absolute configurations of these enantiomers were assigned by ECD calculations at B3LYP/6-31G (d, p) level. The results show that the calculated ECD curve of 7*S*, 8*S*, 7′*R*, 8′*R* agrees well with the experimental one for **2a** (Figure 3). Therefore, the structure of compound **2** was finally assigned and named (+)-ferulasinkin B and (−)-ferulasinkin B, respectively.

Compound **3** was a colorless gum. The relative configuration of **3** was also deduced from the interpretation of ROESY experiments (Figure 2). The correlation of H-8/H-8′ reveals that H-8 and H-8′ are orientated in the same direction. Furthermore, the *J*_7,8_ value of 7.3 Hz suggests that H-7 and H-8 are *threo* configurations. As for the relative configuration at C-7′, ROESY correlations of H-8/H-6′ are observed, indicating the orientation of H-7′, which is *trans* relationship compared to H-8′ corresponding to the *J* value of H-7′ (5.4 Hz). Thus, the relative configuration of **3** was assigned. Likewise, **3** is a racemic mixture indicated by chiral HPLC analysis, which was further separated by chiral HPLC to afford two enantiomers, **3a**, and **3b**, respectively (Appendix A). The absolute configurations of these two enantiomers were eventually determined to be 7*S*, 8*S*, 7′*R*, 8′*S* for **3a** and 7*R*, 8*R*, 7′*S*, 8′*R* for **3b** by comparing the calculated ECD curves with the experimental CD spectra (Figure 3). As mentioned above, compound **3** was assigned and named (+)-ferulasinkin C and (−)-ferulasinkin C, respectively.

Compound **4** was purified as a colorless gum. In the ROESY spectrum of **4**, the cross-peaks of H-8/H-8′ revealed that both H-8 and H-8′ are located in the same direction. Moreover, ROESY correlations of H-7/H-6′ are observed, indicating the orientation of H-7′, which is *cis* relationship compared to H-8′ corresponding to the *J* value of H-7′ (3.9 Hz) (Figure 2). Differing from compounds **1**–**3**, the *J*_7,8_ value of 4.4 Hz suggests that H-7 and H-8 are *erythro* configurations. Compound **4** is a racemic mixture by chiral HPLC analysis. Chiral separation afforded **4a** and **4b** by a Chiralpak AD-H column (Appendix A). Their absolute configurations were assigned as 7*S*, 8*R*, 7′*R*, 8′*R* for **4b** and 7*R*, 8*S*, 7′*S*, 8′*S* for **4a** by comparing its CD spectrum with the experimental one (Figure 3). In this way, the structure of compound **4** was finally identified and named (+)-ferulasinkin D and (−)-ferulasinkin D, respectively.

### 2.2. Biological Evaluation

Breast cancer is a kind of malignant cancer. Especially, triple-negative breast cancer (TNBC) is one of the subgroups of breast cancer which has an incidence of 15–20% and many characteristics such as invasive, resistance, and rapid growth rate [25]. Therefore, discovering some potential molecules which can inhibit cell migration of TNBC is of great importance. Previous research has shown that norlignans possessed an anti-tumor potential [19,20]. Inspired by the traditional medicinal applications of *F. sinkiangensis*, antitumor and wound-healing activities of compounds **1a**/**1b**–**4a**/**4b** by using cell proliferation and cell migration assays in TNBC cells and HUVECs were evaluated [17,26].

The results demonstrated that all compounds did not affect the cell viability of TNBC cell lines (Figure 4A). At the same time, compound **3a** could significantly inhibit the invasion and migration of these two TNBC cell lines. Interestingly, compound **2a** exhibits the inhibitory activity on HCC1806 only (Figure 4B–E). The results of assays on HUVECs demonstrate that compounds **4a** and **4b** could significantly promote cell proliferation (Figure 5A). However, all compounds performed negligible effects on cell migration of HUVECs (Figure 5B,C). In general, cell proliferation, cell mobility, and tube formation assays are used for evaluating the wound-healing potential of the compounds. In the present study, the tube formation test was not continued due to the negligible effects of the compounds on cell migration. Despite this, compounds **4a** and **4b** might be beneficial for wound healing since the cell proliferation promotion of HUVECs is also considered to contribute to wound healing.

As mentioned above, compound **3a** was found to inhibit the invasion and migration of two TNBC cell lines, while **2a** only demonstrated inhibitory effects on the HCC1806 cell line. There are reasons to speculate that the *S* configuration of H-8 and *R* configuration of H-7′ may contribute to the antitumor activities. Furthermore, compared to compound **2a**, the significant inhibitory effects of **3a** may be due to the *trans* relationship between H-7′ and H-8′. As for the wound-healing promotion activities of **4a** and **4b**, these differences may be affected by the *erythro* configuration between H-7 and H-8. Chemically, compounds **1**–**4** share the same chemical skeleton, but they possess different stereo configurations, respectively. These results showed that their biological activities differ greatly, indicating that the changes in the chiral centers of the compounds are crucial to biological activities.

## 3. Experimental Section

### 3.1. General Experimental Procedures

UV spectra and CD spectra were measured on a Jasco J-815 circular dichroism spectrometer (JASCO, Tokyo, Japan). Optical rotations were recorded by using an Anton Paar MCP-100 digital polarimeter (Anton Paar, Graz, Austria). NMR spectra were determined on a Bruker AV-500 and AV-600 spectrometer (Bruker, Karlsruhe, Germany) with TMS as an internal standard. HR-ESI-MS were collected by using a Shimazu LC-20AD AB SCIEX triple TOF X500R MS spectrometer (Shimadzu Corporation, Tokyo, Japan). Column chromatography was performed on silica gel (200–300 mesh, Qingdao Marine Chemical Inc., Qingdao, China), MCI gel CHP 20P (75–150 μm, Mitsubishi Chemical Industries, Tokyo, Japan), RP-18 (40–60 µm, Daiso Co., Tokyo, Japan), YMC gel ODS-A-HG (40–60 μm; YMC Co., Kyoto, Japan), and Sephadex LH-20 (Amersham Pharmacia, Uppsala, Sweden). A Saipuruisi (SEP) chromatograph with a YMC-Pack ODS-A column (250 × 10 mm, i.d., 5 µm) or Kinetex Biphenyl 100A column (250 × 10 mm, i.d., 5 µm) were employed for semi-preparative HPLC. Chiral separation was carried out on a chiral HPLC equipped with a UV detector and a Daicel Chiralpak IC column (250 mm × 4.6 mm, i.d., 5 μm) or a Daicel Chiralpak AD-H column (250 mm × 4.6 mm, i.d., 5 μm) at a flow rate of 1.0 mL/min.

### 3.2. Plant Material

Dried resins of *F. sinkiangensis* K. M. Shen were purchased from Huangyadai Pharmaceutical Co., Ltd., Yunnan, China, in September 2020 and identified by Professor Bin Qiu (Yunnan University of Traditional Chinese Medicine, Kunming, China). A voucher specimen CHYX0670 is deposited at the School of Pharmaceutical Sciences, Shenzhen University, Shenzhen, China.

### 3.3. Extraction and Isolation

Powdered resins of *F. sinkiangensis* (15.0 kg) were extracted with 95% EtOH (3 × 90 L), each for 24 h, at room temperature, and concentrated in vacuo. The crude extract (7.5 kg) was partitioned between H_2_O and EtOAc. The EtOAc portion was combined and evaporated under reduced pressure to give a residue (3 kg), which was subjected to MCI gel CHP 20P eluted with gradient aqueous MeOH (50–100%) to afford ten fractions (Fr.A–Fr.J).

Fr.E (30.5 g) was submitted to a RP-18 column eluted with aqueous MeOH (40–100%) to yield six fractions (Fr.E.1–Fr.E.6). Fr.E.4 (3.4 g) was fractionated by using a YMC gel ODS-A-HG column with gradient aqueous MeOH (30–100%) to generate five fractions (Fr.E.4.1–Fr.E.4.5). Fr.E.4.1 (316 mg) was divided into eight fractions (Fr.E.4.1.1–Fr.E.4.1.8) by preparative thin layer chromatography (PTLC) developed with CH_2_Cl_2_-EtOAc (3:1, *v*/*v*). Fr.E.4.1.7 (60.1 mg) was chromatographed on Sephadex LH-20 (aqueous MeOH, 80%) to produce two parts (Fr.E.4.1.7.1 and Fr.E.4.1.7.2). Fr.E.4.1.7.1 (29.2 mg) was purified by semi-preparative HPLC with aqueous MeOH (28%, 3 mL/min) to afford compound **2** (6.84 mg, t_R_ = 25.6 min) and compound **1** (7.92 mg, t_R_ = 27.5 min). Compound **1** was a racemate that separated by semi-preparative HPLC on a chiral phase equipped with a Daicel Chiralpak IC column (*n*-hexane/iPrOH 65:35, flow rate: 1.0 mL/min) to afford (+)-**1** (t_R_ = 15.8 min, 3.2 mg), **1a**; and (−)-**1** (t_R_ = 20.6 min, 3.0 mg), **1b**. Compound **2** was also isolated as a racemic mixture. This was confirmed by chiral HPLC analysis, which was used with a Daicel Chiralpak AD-H column (*n*-hexane/EtOH 55:45, flow rate: 1.0 mL/min) to afford (+)-**2** (t_R_ = 11.3 min, 3.0 mg), **2a**; and (−)-**2** (t_R_ = 15.7 min, 2.8 mg), **2b**.

In order to further purify Fr.E.5 (10.2 g), it was submitted to a YMC gel ODS-A-HG column eluted with aqueous MeOH (35–100%) to yield eight parts (Fr.E.5.1–Fr.E.5.8). Fr.E.5.2 (237.6 mg) was subjected to Sephadex LH-20 (aqueous MeOH, 80%) to afford three fractions (Fr.E.5.2.1–Fr.E.5.2.3). Fr.E.5.2.2 (150.3 mg) was applied to a silica gel column eluted with petroleum ether-Me_2_CO (5:1–0:1, *v*/*v*) to afford eight parts (Fr.E.5.2.2.1–Fr.E.5.2.2.8). Among them, Fr.E.5.2.2.6 (80.4 mg) was divided into six fractions (Fr.E.5.2.2.6.1–Fr.E.5.2.2.6.6) by PTLC developed with CH_2_Cl_2_-Me_2_CO (3:2). Fr.E.5.2.2.6.3 (34 mg) was purified by using semi-preparative HPLC eluted with aqueous MeOH (28%, 3 mL/min) to afford compounds **4** (3.5 mg, t_R_ = 24.0 min) and **3** (7.9 mg, t_R_ = 32.5 min). Racemate **3** was separated by semipreparative HPLC on a chiral phase equipped with a Daicel Chiralpak IC column (*n*-hexane/EtOH 75:25, flow rate: 1.0 mL/min) to obtain (+)-**3** (t_R_ = 19.9 min, 3.03 mg), **3a**; and (−)-**3** (t_R_ = 22.8 min, 3.28 mg), **3b**. Compound **4**, as a racemic mixture, was further purified by semi-preparative HPLC on a chiral phase equipped with a Daicel Chiralpak AD-H column (*n*-hexane/EtOH 68:32, flow rate: 1.0 mL/min) to afford (+)-**4** (t_R_ = 12.0 min, 1.12 mg), **4a**; and (−)-**4** (t_R_ = 15.6 min, 1.29 mg), **4b**, respectively.

### 3.4. Compound Characterization Data

Ferulasinkin A (**1**): colorless gum; UV (MeOH) λ_max_ (logε) 201 (3.97), 216 (2.97), 230 (3.11), 282 (2.63) nm; {[α]^20^_D_ −19.3 (c 0.05, MeOH); CD (MeOH) Δε_203_ −16.08, Δε_219_ −0.87, Δε_233_ −2.43; **1a**}; {[α]^20^_D_ +26.0 (c 0.05, MeOH); CD (MeOH) Δε_200_ +20.86, Δε_216_ +1.03, Δε_223_ +2.17; **1b**}; HR-ESI-MS (*m/z* 361.1293 [M − H]^−^, calcd for 361.1278, C_19_H_22_O_7_); ^1^H- and ^13^C-NMR data, see Table 1.

Ferulasinkin B (**2**): colorless gum; UV (MeOH) λ_max_ (logε) 202 (3.93), 217 (3.01), 230 (3.13), 280 (2.69) nm; {[α]^20^_D_ −19.3 (c 0.05, MeOH); CD (MeOH) Δε_209_ +2.90, Δε_221_ −0.02, Δε_233_ –0.83; **2a**}; {[α]^20^_D_ +23.3 (c 0.05, MeOH); CD (MeOH) Δε_209_ −5.94, Δε_221_ +0.17, Δε_237_ +0.58; **2b**}; HR-ESI-MS (*m/z* 361.1293 [M − H]^−^, calcd for 361.1278, C_19_H_22_O_7_); ^1^H- and ^13^C-NMR data, see Table 1.

Ferulasinkin C (**3**): colorless gum; UV (MeOH) λ_max_ (logε) 202 (4.18), 216 (3.21), 231 (3.35), 282 (2.89) nm; {[α]^20^_D_ −21.0 (c 0.05, MeOH); CD (MeOH) Δε_204_ −15.68, Δε_219_ −1.60, Δε_232_ −2.82; **3a**}; {[α]^20^_D_ +19.3 (c 0.05, MeOH); CD (MeOH) Δε_200_ +6.85, Δε_218_ +0.54, Δε_233_ +0.87; **3b**}; HR-ESI-MS (*m/z* 361.1293 [M − H]^−^, calcd for 361.1280, C_19_H_22_O_7_); ^1^H- and ^13^C-NMR data, see Table 2.

**Table 2 molecules-27-03907-t002:** ^1^H (600 MHz) and ^13^C (150 MHz) NMR data of **3** and **4** in MeOD (*δ* in ppm, *J* in Hz).

No.	3		4	
	*δ* _H_	*δ* _C_	*δ* _H_	*δ* _C_
1		134.1, s		134.6, s
2	7.02 (d, 1.9)	111.8, d	7.04 (d, 1.9)	111.6, d
3		148.9, s		148.9, s
4		147.0, s		147.0, s
5	6.79 (t, 8.1)	115.9, d	6.78 (d, 8.0)	115.8, d
6	6.86 (dd, 8.1, 1.9)	121.1, d	6.86 (dd, 8.0, 1.9)	120.6, d
7	4.64 (d, 7.3)	78.1, d	4.77 (d, 4.4)	75.9, d
8	4.39 (td-like, 7.2, 6.2)	83.6, d	4.39 (dt, 6.3, 4.4)	83.7, d
9	Ha: 2.04 (dt, 13.1, 7.2); Hb: 1.72 (dt, 13.1, 6.2)	37.5, t	Ha: 2.22 (ddd, 13.0, 8.7, 6.3); Hb: 1.80 (ddd, 13.0, 6.3, 3.9)	36.0, t
1′		133.7, s		133.9, s
2′	6.95 (d, 1.9)	110.6, d	7.02 (d, 1.9)	110.9, d
3′		148.9, s		148.7, s
4′		147.3, s		146.9, s
5′	6.76 (d, 8.1)	115.9, d	6.74 (d, 8.0)	115.7, d
6′	6.82 (dd, 8.1, 1.9)	119.7, d	6.80 (dd, 8.0, 1.9)	120.0, d
7′	4.68 (d, 5.4)	87.7, d	4.60 (d, 3.9)	89.5, d
8′	4.12 (td-like, 6.2, 5.5)	79.1, d	4.05 (dt, 7.0, 3.9)	79.1, d
3-OCH_3_	3.87 (s)	56.3, q	3.83 (s)	56.3, q
3′-OCH_3_	3.85 (s)	56.3, q	3.83 (s)	56.3, q

Ferulasinkin D (**4**): colorless gum; UV (MeOH) λ_max_ (logε) 201 (3.76), 215 (2.76), 231 (2.86), 280 (2.41) nm; {[α]^20^_D_ −6.0 (c 0.04, MeOH); CD (MeOH) Δε_208_ +0.85, Δε_225_ +0.27, Δε_238_ +0.57; **4a**}; {[α]^20^_D_ +8.6 (c 0.04, MeOH); CD (MeOH) Δε_209_ −5.41, Δε_223_ −1.57, Δε_234_ −2.70; **4b**}; HR-ESI-MS (*m/z* 361.1293 [M − H]^–^, calcd for 361.1278, C_19_H_22_O_7_); ^1^H- and ^13^C-NMR data see Table 2.

### 3.5. ECD Calculations

The CONFLEX searches based on molecular mechanics with the Molecular Merck force field (MMFF) were performed for compounds **1**–**4**, which gave several stable conformers for each compound. Furthermore, the selected conformers of each compound with the lowest energy were optimized and calculated at the B3LYP/6-31G (d, p) level in a Gaussian 09 program (Gaussian Inc., Wallingford, CT, USA) [27]. Finally, the comparison of ECD spectra was obtained by using SpecDis 1.62 and Origin 2021 programs (OriginLab, Northampton, MA, USA).

### 3.6. Inhibitory Effects on TNBC Cells

The human TNBC cell lines HCC1806 and MDA-MB231 (Cell Bank of Chinese Academy of Sciences, Shanghai, China) were cultured in 10 cm dishes and maintained in Dulbecco’s modified Eagle’s medium (DMEM) containing 10% (*v*/*v*) fetal bovine serum (FBS) with 100 U/mL penicillin and incubated at 37 °C in a 5% CO_2_, 95% humidified atmosphere. Cellular metabolic activity was measured with a Cell Count Kit-8 (CCK-8) (Beyotime, Shanghai, China) assay kit after a 40 μM compound or DMSO treatment. The cells were seeded in 96-well plates at a density of 3–5 × 10^3^ cells/well for 1 day and then incubated in 2% FBS media with or without compounds for 48 h. Subsequently, 10 μL of 5 mg/mL CCK-8 solution was added to each well and incubated at 37 °C for an additional 4 h. The absorbance of each well was determined at 450 nm with a microplate reader (BioTek, Winooski, VT, USA). The confluent TNBC cells in 96 cultured plates were scratched vertically by using Scratch Assay Starter Kit (BioTek, Winooski, VT, USA) and incubated with compounds (40 μM) and DMSO as vehicles in DMEM containing 0.5% FBS for 48 h. Images of the wounded region were analyzed by CYTATION1 (BioTek, Winooski, VT, USA) after scratches occurred. Cell migration of this region was monitored at 0 h, 24 h, and 48 h (T0, T24, and T48), respectively. The migration width of T24 (width at T0 − width at T24) and T48 (width at T0 − width at T48) were recorded.

### 3.7. Promoting Effects on HUVECs

HUVECs (iCell Bioscience Inc., Shanghai, China) were cultured in 10 cm dishes and maintained in endothelial cell medium (ECM) (Sciencell, San Diego, CA, USA) containing 5% FBS, 1% penicillin, 1% streptomycin, and 1% endothelial cell growth supplement (ECGS) (Sciencell, San Diego, CA, USA), and incubated at 37 °C in a 5% CO_2_, 95% humidified atmosphere. Cellular metabolic activity was also measured with a CCK-8 assay kit after a 40 μM compound or DMSO treatment. These growing cells were seeded in 96-well plates at a density of 3–5 × 10^3^ cells/well with completed ECM. After cell adhesion and cellular morphology restoration, they were starved for 12–16 h, then incubated with or without compounds for 48 h. At this time, 90 μL fresh ECM and 10 μL of 5 mg/mL CCK-8 solution were added to each well and incubated at 37 °C for 4 h. A microplate reader was used to determine the absorbance of each well at 450 nm. To conduct a wound-healing assay, cells were pre-treated with serum-free medium for 12–16 h until they grew to form a monolayer in 96 cultured plates. Then these cells were scratched vertically by using Scratch Assay Starter Kit and incubated with compounds (40 μM) and DMSO as vehicles in ECM containing 0.5% FBS for 24 h. Images of the wounded region were analyzed by CYTATION1 after scratches occurred. Cell migration of this region was monitored at 0 h, 24 h, and 48 h (T0, T24, and T48), respectively. The migration width of T24 (width at T0 − width at T24) and T48 (width at T0 − width at T48) were recorded.

### 3.8. Statistical Analysis

All biological experiments in this study were carried out in triplicate. The data are means ±SEM. Graphpad Prism 7 (GraphPad Software, San Diego, CA, USA) was used to conduct statistical analyses, which included a Student’s *t*-test and a one-way ANOVA test. When * *p* ≤ 0.05, ** *p* ≤ 0.01, *** *p* ≤ 0.001 and **** *p* ≤ 0.0001, differences were considered significant.

## 4. Conclusions

In summary, four new norlignans, ferulasinkins A–D, were isolated as racemic mixtures from the resins of *F. sinkiangensis*. Their respective enantiomers were obtained via chiral HPLC separation. Compounds **2a** and **3a** could significantly inhibit the invasion and migration of TNBC cell lines. Additionally, compounds **4a** and **4b** could promote cell proliferation of HUVECs, showing their wound-healing effects. These findings help us to gain a deep insight into the chemical profiling and biological properties of *F. sinkiangensis* resins.

## Figures and Tables

**Figure 1 molecules-27-03907-f001:**
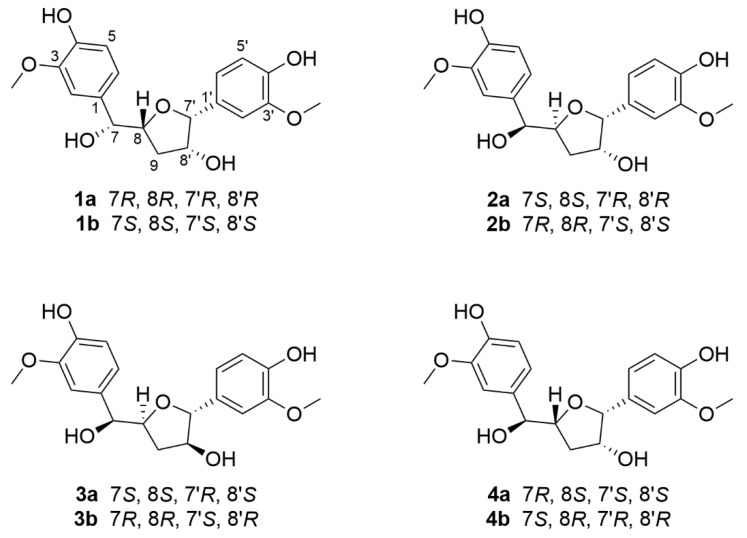
The structures of compounds **1**–**4** from *F.*
*sinkiangensis*.

**Figure 2 molecules-27-03907-f002:**
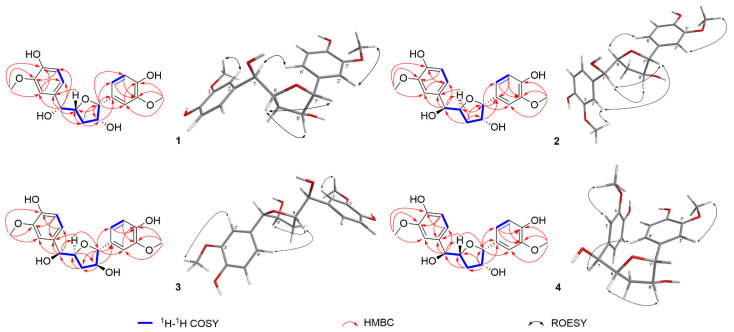
Key 2D NMR correlations of compounds **1**–**4**.

**Figure 3 molecules-27-03907-f003:**
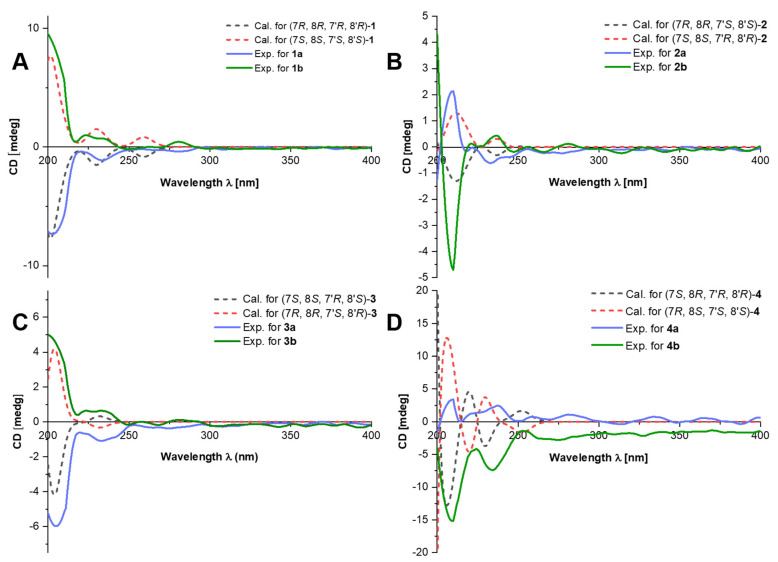
Comparison of the calculated ECD and experimental spectra in MeOH. (**A**) The calculated ECD spectra of (7*R*, 8*R*, 7′*R*, 8′*R*)-**1** and (7*S*, 8*S*, 7′*S*, 8′*S*)-**1** at B3LYP/6-31G level, σ = 0.30 eV; shift = 3 nm. (**B**) The calculated ECD spectra of (7*R*, 8*R*, 7′*S*, 8′*S*)-**2** and (7*S*, 8*S*, 7′*R*, 8′*R*)-**2** at B3LYP/6-31G level, σ = 0.22 eV; shift = 11 nm. (**C**) The calculated ECD spectra of (7*S*, 8*S*, 7′*R*, 8′*S*)-**3** and (7*R*, 8*R*, 7′*S*, 8′*R*)-**3** at B3LYP/6-31G level, σ = 0.25 eV; shift = 18 nm. (**D**) The calculated ECD spectra of (7*S*, 8*R*, 7′*R*, 8′*R)-***4** and (7*R*, 8*S*, 7′*S*, 8′*S)-***4** at B3LYP/6-31G level, σ = 0.20 eV; shift = 0 nm.

**Figure 4 molecules-27-03907-f004:**
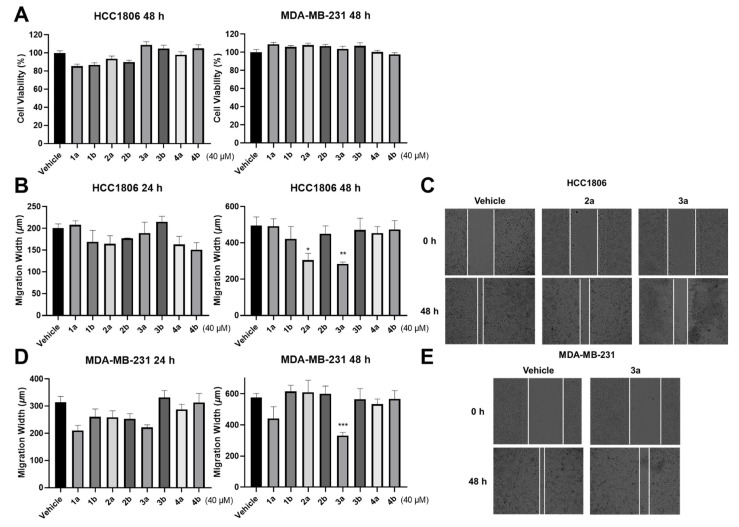
Inhibitory effects of all optically active compounds on TNBC cell lines. (**A**) Treatment of compounds on HCC1806 and MDA-MB-231 cell lines did not affect cell viability. (**B**–**E**) Effects of compounds on migration situation of TNBC cell lines were evaluated by wound-healing experiments (40 μM compounds or DMSO). The data are means ± standard error of the mean (SEM); *n* ≥ 3. * *p* ≤ 0.05, ** *p* ≤ 0.01 and *** *p* ≤ 0.001 versus the vehicle group (one-way ANOVA).

**Figure 5 molecules-27-03907-f005:**
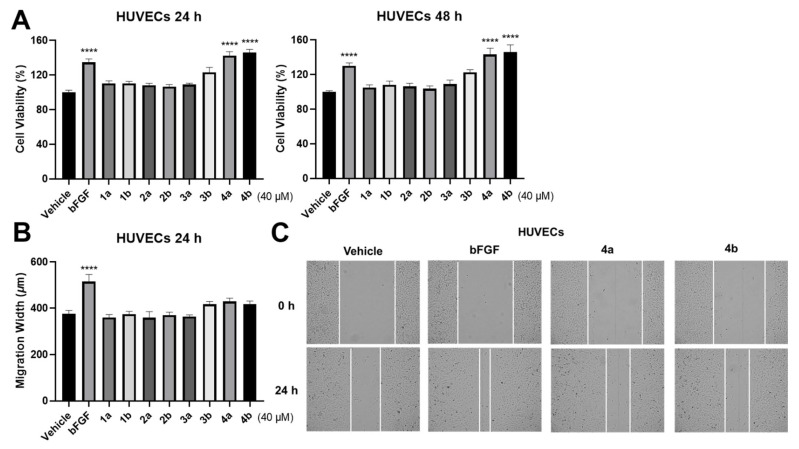
Promoting effects of all optically active compounds on HUVECs. (**A**) Effects of compounds on proliferation situation of HUVECs were evaluated by CCK-8 assay (40 μM compounds or DMSO). (**B**,**C**) Treatment of compounds on HUVECs did not affect cell migration. The data are means ± SEM; *n* ≥ 3. **** *p* ≤ 0.0001 versus the vehicle group (one-way ANOVA). Basic fibroblast growth factor (bFGF) (100 ng/mL) was used as a positive control.

**Table 1 molecules-27-03907-t001:** ^1^H (600 MHz) and ^13^C (150 MHz) NMR data of **1** and **2** in MeOD (*δ* in ppm, *J* in Hz).

No.	1		2	
	*δ* _H_	*δ* _C_	*δ* _H_	*δ* _C_
1		134.2, s		133.9, s
2	7.01 (d, 1.9)	111.7, d	7.03 (d, 1.9)	111.7, d
3		148.8, s		148.6, s
4		147.0, s		147.3, s
5	6.78 (d, 8.1)	115.8, d	6.78 (d, 8.1)	115.0, d
6	6.89 (dd, 8.1, 1.9)	121.0, d	6.85 (dd, 8.1, 1.9)	121.0, d
7	4.61 (d, 7.0)	78.1, d	4.53 (d, 7.7)	78.1, d
8	4.40 (dt, 7.0, 6.5)	83.8, d	4.57 (m)	83.3, d
9	Ha: 1.91 (ddd, 13.1, 6.1, 6.5); Hb: 1.57 (ddd, 13.1, 6.6, 3.6)	37.6, t	Ha: 1.99 (ddd, 13.4, 9.3, 4.6); Hb: 1.75 (ddd, 13.4, 6.1, 1.0)	38.7, t
1′		133.9, s		130.7, s
2′	6.99 (d, 1.9)	110.8, d	7.05 (d, 1.9)	112.3, d
3′		148.9, s		148.9, s
4′		147.3, s		146.8, s
5′	6.75 (d, 8.1)	115.9, d	6.76 (d, 8.1)	115.0, d
6′	6.84 (dd, 8.1, 1.9)	119.8, d	6.81 (dd, 8.1, 1.9)	121.0, d
7′	4.68 (d, 3.6)	89.8, d	4.87 (d, 3.2)	86.2, d
8′	4.02 (td, 6.6, 3.6)	79.2, d	4.22 (t, 3.8)	75.1, d
3-OCH_3_	3.84 (s)	56.3, q	3.86 (s)	56.3, q
3′-OCH_3_	3.86 (s)	56.3, q	3.86 (s)	56.3, q

## Data Availability

Data is contained within the article or Appendix A.

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
