# Peer review of "Racemic Norlignans as Diastereoisomers from Ferula sinkiangensis Resins with Antitumor and Wound-Healing Promotion Activities"

_molecules, 2022, doi:10.3390/molecules27123907_

Round 1

Reviewer 1 Report

The manuscript discussed structure elucidation of four new norlignans namely: Ferulasinkines A-D (1-4), four new norlignans  isolated from the resins of Ferula sinkiangensis

There is minor language mistakes should be revised (e.g .Apiaceous and the name of the plant should be written in italic).

Author Response

Point 1: There is minor language mistakes should be revised (e.g .Apiaceous and the name of the plant should be written in italic)

Response 1Thank you for your advice. We have revised in the whole MS.

Reviewer 2 Report

The MS entitled ``Racemic Norlignans as Diastereoisomers from Ferula sinkiangensis Resins with Antitumor and Wound Healing Promotion Activities`` cannot be published in its current form. The following issues should be addressed.

English should be carefully revised through the whole MS.

Abstract

The plant`s family name should be added in the abstract and keywords. Also, Ferulasinkines should be added in the keywords

-The abstract needs improvement. More about the biological activity should be added including type of assay, results for both compounds and positive controls.

_ suggested for the authors to change compounds names to Ferulasinkins A-D instead of Ferulasinkines A-D to avoid confusion with alkaloids suffix (ine).

-The genus name and F. sinkiangensis should be italized and check through the whole MS.

-Apiaceous should be corrected to Apiaceae.

-Authors should avoid the use of ``We``.

-Introduction is too short, authors should write about norlignans and their structures, Are norlignane reported from any of Ferula species? This should be discussed in the introduction with related references.

-IR and UV data of the compounds should be added and discussed in the compounds` discussion.

-No need to write the number of proton, multiplicity, and J values for each signal as in ``(δH 7.01, 1H, d, J = 1.9 Hz; δH 6.99, 1H, d,…..` this is already present in the table, just use the chemical shifts

-The H-H COSY and HMBC correlations that confirm the presence of ABX systems or trisubstituted aromatic moieties should be discussed in detail, also reference should be added.

-this sentence ``Its molecular formula C19H22O7 (nine degrees of unsaturation) ……`` need rephrasing.

-Authors should mention the HMBC correlations that confirm the location of methoxy groups.

- Ferulasinkine A should be changed to ferulasinkin A, all names should be corrected accordingly

- To enrich the value of the MS, the Possible biosynthetic pathway of these metabolites should be added and discussed.

- Biological evaluation part needs improvement. Authors should discuss why they select to test the antitumor and wound healing potential of these metabolites and why they use these cell lines specifically. Are the data the got agree with the previously reported data for related compounds. All these should be included in the MS with the related references. Structure activity relationship should be discussed with highlighting the effect of chiral centres on the activity.

Author Response

Dear reviewer,

We have improved and made some changes in whole MS according to your advice. The response is included in the uploaded Word file.

We sincerely hope that the revision and explanation will meet with approval. Thank you once again for your valuable comments and suggestions.

We wish your positive feedback.

Have a nice day.

Reviewer 3 Report

The article concerns the isolation and structural elucidation of four racemic norglycans from the resins of Ferula sinkiangensis, ferulasinkines A–D. All of them were firstly isolated as racemic corresponding mixtures. The mixtures were subdivided on (+) and (-) enantiomers using chiral HPLC. The planar structures were elucidated by 2D NMR and HR ESIMS, the relative stereochemistry was determined by the ROESY and HMBC correlations and absolute configurations were determined using CD-spectroscopy and computation methods. The isolated optically active natural products were tested for their antitumor and wound healing properties. All the substances were no cytotoxic but two of them significantly delayed invasion and migration of the tumor cells. Two substances showed potential wound healing effect. The work seems to be a significant contribution into the phytochemical profiling of F. sinkangensis.

The article is very well written, and will be very interesting for the readers. Only two minor corrections are necessary. I strictly recommend to duplicate the conditions of the NMR spectra obtaining (working frequencies for 13C and 1H-NMR spectra) in the experimental part also and add the experimental conditions for obtaining of HRESIMS.

The article may be published after very minor corrections without any further excessive correspondence.

Author Response

Point 1: Only two minor corrections are necessary. I strictly recommend to duplicate the conditions of the NMR spectra obtaining (working frequencies for 13C and 1H-NMR spectra) in the experimental part also and add the experimental conditions for obtaining of HRESIMS.

Response 1: Thank you for your advice. It's really true as you suggest that the experimental conditions of the NMR and HRESIMS spectra obtaining are of great importance. Actually, these conditions have been included in the main text (Tab. 1-2 and characterization data) and supplementary material. As per your valuable advice, we have added this information in the main text and supplementary material again (as shown on page 2, line 66; page 4, line 114, 125, 139; each figure legends of the NMR spectra in supplementary material). Our previous published articles also use this way to demonstrate HRESIMS spectra in the supplementary material (Yan, Y.M.; Chen, H.; Chen, W.L.; Wang, D.W.; Liao, L.; Lu, Q.; Cheng, Y.X. Alkyl-modified nucleobases with 6/5/7/5 ring systems from the insect Cyclopelta parva. Org. Chem. Front. 2021, 9, 75-80. doi:10.1039/d1qo01603b) (Huang, X.L.; Wang, D.W.; Liu, Y.Q.; Cheng, Y.X. Diterpenoids from Blumea balsamifera and their anti-inflammatory activities. Molecules. 2022, 27. doi:10.3390/molecules27092890).

We sincerely hope that the revision will meet with approval. Once again, thank you very much for your comments and suggestions. Have a nice day.

Round 2

Reviewer 2 Report

No further comments.